# One-Pot NBS-Promoted Synthesis of Imidazoles and Thiazoles from Ethylarenes in Water

**DOI:** 10.3390/molecules24050893

**Published:** 2019-03-04

**Authors:** Liang Chen, Huajian Zhu, Jiang Wang, Hong Liu

**Affiliations:** 1School of Pharmacy, China Pharmaceutical University, Nanjing 210009, Jiangsu, China; zhulinfengcl@163.com (L.C.); cpuzhj@163.com (H.Z.); 2State Key Laboratory of Drug Research and CAS Key Laboratory of Receptor Research, Shanghai Institute of Materia Medica, Chinese Academy of Sciences, Shanghai 201203, Shanghai, China

**Keywords:** NBS, nitrogen-containing heterocycle, water, one-pot

## Abstract

A facile and eco-friendly method has been developed for the synthesis of imidazoles and thiazoles from ethylarenes in water. The reaction proceeds via in situ formation of α-bromoketone using NBS as a bromine source as well as an oxidant, followed by trapping with suitable nucleophiles to provide the corresponding products in good yields under metal-free conditions.

## 1. Introduction

Nitrogen-containing heterocycles are found in many biologically active synthetic targets, including natural products and designed pharmaceuticals [1,2,3]. Therefore, the construction and functionalization of nitrogen-containing heterocycles has attracted considerable attention from synthetic chemists. Imidazoles and thiazoles are of particular interest, as these building blocks have been incorporated into a number of bioactive compounds including zolpidem, miroprofen, amiphenazole, zolimidine, YM-11124, and CICTO (Figure 1) [4,5,6,7,8,9,10,11,12,13,14,15,16]. These compounds exhibit a plethora of biological properties displayed over a broad range of therapeutic classes, including antibacterial, antifungal, antiviral, antiulcer, anti-inflammatory, β-amyloid formation inhibitory, immunosuppressive, GABA receptor agonist, cardiotonic agent, and nonpeptide B_2_ receptor antagonist effects [17,18,19,20,21]. Various synthetic methods have been reported for the construction of these motifs such as C-H amination, oxidative cyclization, multi-component reaction, hydroamination, and tandem processes from various starting materials [22,23,24,25,26]. Traditionally, procedures for these reactions required the use of metals and catalysts in various organic solvents, which makes the sequence longer and increases waste production. For example, Toste et al. reported a dichloro(2-pyridinecarboxylato)-gold [PicAuCl_2_]-catalyzed reaction of 2-aminopyridine *N*-oxide and alkynes in dichloromethane for the synthesis of imidazo[1,2-a]pyridines. The reaction needs an expensive gold catalyst and an acid additive, which increases costs and creates pollution [27]. Hence, the development of new methodologies for the synthesis of these compounds continues to attract the interest of academic and industrial researchers. 

As a direct and efficient approach to the synthesis of these imidazoles and thiazoles, α-halo-ketones have reacted with suitable nitrogen nucleophiles and bases in various organic solvents (Scheme 1a) [28,29]. Although this method is suitable for certain synthetic conditions, sometimes, however, these procedures have one or more disadvantages such as the use of hazardous organic solvents, long reaction times, use of stoichiometric and even excess amounts of reagents, etc. To overcome these limitations, Mahesh and co-workers recently reported a two-step approach to synthesize these heterocycles from alkenes, which involves preparing α-bromoketones by reaction of olefins with NBS in water, and these α-bromoketones were treated with suitable nucleophiles to give a diverse range of imidazoles and thiazoles (Scheme 1b) [30]. This method is only applied in synthesis of some specific nitrogen-containing heterocycles, but, other heterocycles are not mentioned. α-Bromoacetophenone was also reported by Shimokawa et al. to be the important intermediate dealing with ethylarenes with NBS in a mixture of ethyl acetate and water [31]. In this report, a variety of ethylarenes were converted into the corresponding primary aromatic amides. Inspired by this work, we envisaged the possibility to synthesize various imidazoles and thiazoles from ethylarenes.

Here, we report an NBS-promoted, one-pot method for the construction of various imidazoles and thiazoles from ethylarenes in water as a solvent (Scheme 1c). 

NBS plays a dual role of both a safe bromine source and an oxidant, and the reaction was carried out with water, which also acts as the oxygen source for the in situ preparation of α-bromoketones. Water is an economical, safe, and environmentally benign solvent, and, therefore, its use as a solvent for organic reactions is a very attractive option [32,33]. This protocol presents an operationally simple, rapid, and environmentally friendly strategy for enriching a complex nitrogen heterocycle library.

## 2. Results and Discussion

### 2.1. Optimization of Reaction Conditions for Synthesis of Imi-azo[1,2-a]pyridine ***3a***

Our study was initiated by treating ethylbenzene (**1a**, 1 mmol) as model substrate with NBS (3.5 equiv.) in the presence of AIBN (10 mol %) in a mixture of ethyl acetate:water (5:1, 6 mL) at 65 °C for 1.5 h, followed by addition of 2-aminopyridine (**2a**) at 80 °C to give the desired imidazo[1,2-a]pyridine (**3a**) in 38% yield (Table 1, entry 1). Interestingly, when NBS was replaced with *N*-chlorosuccinimide (NCS) or *N*-iodosuccinimide (NIS), formation of **3a** was not observed (Table 1, entries 2 and 3), whereas with *N*-bromophthalimide (BrNPhth), compound **3a** was obtained in 34% yield (Table 1, entry 4). 

No improvement in the yield was observed by using acetonitrile, acetone, and 1,4-dioxane instead of ethyl acetate (Table 1, entries 5–7). A series of screening reactions to optimize the amount of oxidant, solvent system and reaction temperature was also performed, which had no good effect (Appendix A, see the Appendix A for details). After extensive experimentation, it was found that the presence of ethyl acetate in the reaction mixture was detrimental to the second reaction step. Hence, the solvent of second reaction step was replaced with water, ethanol, and acetone [34,35]. Water was found to be the best solvent, furnishing imidazo[1,2-a]pyridine (**3a**) in 67% yield (Table 1, entries 8–10). Further optimization of the reaction conditions was made by adding sodium carbonate (Table 1, entries 11 and 12). In these cases, the yield of target product was improved to 78%. It is presumed that the sodium carbonate neutralizes the acid formed in the reaction, which accelerates the intramolecular cyclization of the second step. Bringing the temperature to reflux did not have any noteworthy effect on the yield of **3a** (Table 1, entry 13).

### 2.2. Substrate Scope for the Imidazo[1,2-a]pyridines

With optimized conditions in hand, the reactions of different 2-aminopyridines with various ethylarenes were investigated to explore the scope and generality of this method for the synthesis of different imidazo[1,2-a]pyridines (Table 2). The reaction of 2-aminopyridine (**2a**) with ethylarenes containing electron-withdrawing groups on the aromatic ring such as halogen, trifluoromethyl, methylsulfonyl, and cyano, provided the corresponding products **3b**–**3h** smoothly in medium to good yields (32% and 65%). Ethylbenzenes with electron-rich groups, such as methyl and methoxy, also afforded the corresponding imidazo[1,2-a]pyridines **3i** and **3j** in very good yield (68% and 59%). The same treatment of 1-ethylnaphthalene gave the desired product **3k** in good yield (65%). Furthermore, electron-rich and electron-withdrawing groups in the 2-aminopyridines were well tolerated. The results revealed that electron-rich groups such as methyl and methoxyl afforded the corresponding products in very good yield (yields 75–80%, **3l**–**3p**). However, introduction of the electron-withdrawing groups such as halogen, trifluoromethyl, and cyano afforded the corresponding products in lower yields (yields 56–71%, **3q**–**3u**).

### 2.3. Substrate Scope for Diverse Imidazoles and Thiazoles 

Encouraged by the results above, we shifted to investigate the scope of other nucleophiles. As shown in Figure 2, a range of nucleophiles were suitable for this reaction. These substrates **4a**–**4g** were effective to give corresponding products **5a**–**5g** in moderate to high yields (60–91%). It is worth noting that valuable and more-complex substrates were also suitable for this reaction to afford corresponding products, which is extremely useful for enriching a complex nitrogen-containing heterocycle library.

### 2.4. Gram-Scale Preparation and Practice Application 

To evaluate the efficiency and potential for practical applications of our method, a scale-up experiment was carried out under the standard conditions (Scheme 2a). As a result, preparation of **3a** on a gram scale (2.95 g) was carried out, giving 76% yield. The compatibility of this reaction encouraged us to start the study of large-scale synthesis of zolimidine, which is a gastroprotective drug previously used for peptic ulcers and gastroesophageal reflux disease [36,37,38]. Starting with 1-bromo-4-ethylbenzene (**1c**) the intermediate product **1h** was obtained on the basis of a literature procedure [39]. Then, according to the standard conditions, the target product **3h** was obtained in 40% yield in total (Scheme 2b).

### 2.5. Mechanism

A series of mechanistic experiments were performed to shed more light on the reaction mechanism, as shown in Scheme 3.

When ethylbenzene (**1a**) was treated with NBS (3.5 equiv.) in the presence of AIBN (10 mol%) in a mixture of ethyl acetate and water (5:1) at 65 °C for 1.5 h, 2-bromoacetophenone (**1aa**) was obtained in 86% yield. At the same time, α,α-dibromoethybenzene (**1ab**) and acetophenone (**1ac**) were formed in 7% and 3% yields, respectively (Scheme 3b). 2-Phenylimidazo[1,2-a]pyridine (**3a**) was obtained in 95% yield when 2-bromoacetophenone (**1aa**) was directly treated with pyridin-2-amine (**2a**) at 80 °C for 2 h (Scheme 3c). α,α-Dibromoethybenzene (**1ab**) was treated with NBS (1.5 equiv.) in the presence of AIBN (10 mol %) in a mixture of ethyl acetate and water (5:1) at 65 °C for 1.5 h, followed by the reaction with **2a** at 80 °C for 2 h to give **3a** in 71% yields (Scheme 3d). Besides, when acetophenone (**1ac**) was treated with NBS (1 equiv.) at 60 °C for 1 h, followed by the reaction with **2a** at 80 °C for 2 h, **3a** was obtained in 86% yield (Scheme 3e). As a result, α,α-dibromoethybenzene (**1ab**) and acetophenone (**1ac**) could also be converted into **3a** using the present reaction procedure and conditions, which contributes to improve the yield of this reaction. On the basis of mechanistic experiments and a literature survey [24,40], a plausible pathway for the reaction has been proposed, as shown in Scheme 4, taking the reaction of ethylbenzene (**1a**) and 2-aminopyridine (**2a**) for the synthesis of imidazo[1,2-a]pyridine (**3a**) as an example. α-Bromoethybenzene is formed in the reaction of ethylbenzene (**1a**) with bromine atoms formed from NBS (Wohl-Ziegler reaction) [41]. α,α-Dibromoethybenzene (**1ab**) is the main product of the second Wohl-Ziegler reaction of α-bromoethylarene. The hydrolysis of α,α-dibromoethybenzene (**1ab**) in a mixture of ethyl acetate and water under warming conditions proceeds to give acetophenone (**1ac**). Meanwhile, the hydrolysis of α-bromoethybenzene may occur to form α-hydroxyethybenzene as a minor product. Acetophenone (**1ac**) can also be easily obtained from α-hydroxyethybenzene oxidized by NBS. Once acetophenone (**1ac**) is formed, it smoothly reacts with NBS to form the important intermediate 2-bromo-acetophenone (**1aa**) which is isolated by column chromatography and confirmed by comparison of NMR with literature. Finally, treatment of 2-bromoacetophenone (**1aa**) with **2a** afforded the desired product **3a** by cyclization.

## 3. Materials and Methods 

### 3.1. General Information 

The reagents (chemicals) were purchased from commercial sources, and used without further purification. Analytical thin layer chromatography (TLC) was performed on HSGF 254 (0.15–0.2 mm thickness) plates. All products were characterized by their NMR and MS spectra. The ^1^H- (500 MHz) and ^13^C-NMR (125 MHz) spectra were recorded in deuterochloroform (CDCl_3_) on Bruker Avance III spectrometer (Billerica, MA, USA). Chemical shifts were reported in parts per million (ppm, δ) downfield from tetramethylsilane. Proton coupling patterns are described as singlet (s), doublet (d), triplet (t), quartet (q), multiplet (m). Low- and high-resolution mass spectra (LRMS and HRMS) were measured on Agilent 1260 Infinity II and Agilent 1290-6545 UHPLC-QTOF respectively (Palo Alto, CA, USA).

### 3.2. Experimental Part Method

#### 3.2.1. General Procedure for the Synthesis of Nitrogen-Containing Heterocycles

Synthesis of **3a** is representative. To a solution of ethylbenzene (**1a**, 1 mmol, 107 mg) in ethyl acetate:water (5:1, 6 mL) were added NBS (3.5 mmol, 628 mg) and AIBN (0.1 mmol, 16.5 mg) at room temperature, and the mixture was stirred at 65 °C for 1.5 h. The mixture was concentrated to dryness and then dissolved in water (5 mL), followed by reaction with 2-aminopyridine (**2a**, 1.2 mmol, 114 mg) and sodium carbonate (5 mmol, 534 mg) for 2 h at 80 °C. After completion of the reaction (as indicated by TLC), the crude product was extracted with ethyl acetate (3 × 10 mL). The combined organic layer was dried over anhydrous Na_2_SO_4_ and concentrated in vacuo. The crude product was purified by silica gel column chromatography (PE/EA = 8/1–4/1, *v*/*v*) to give **3a** (78% yield) as a white solid.

#### 3.2.2. Procedure for the Synthesis of 1-Ethyl-4-(methylsulfonyl)benzene (**1h**)

1-Bromo-4-ethylbenzene (**1c**, 10 mmol), DMSO (50 mL), acetyl acetone (10 mmol), Cu_2_O (1 mmol), and *t*-BuOK (45 mmol) were added into the reactor in turn. The reaction was carried out at reflux (100 °C) under an air atmosphere for 20 h. After the reaction was finished, the mixture was diluted with the saturated NaCl aqueous solution and extracted with ethyl acetate (3 × 40 mL). The organic layers were combined, and then dried with anhydrous Na_2_SO_4_. After the solvent was removed by rotovapor, the product was purified by column chromatograph (PE/EA = 5/1, *v*/*v*).

### 3.3. Product Characterization 

*2-Bromo-1-phenylethan-1-one* (**1aa**) [42]: white solid, ^1^H-NMR (CDCl_3_) δ 8.04–7.95 (m, 2H), 7.65–7.59 (m, 1H), 7.53–7.47 (m, 2H), 4.46 (s, 2H). ^13^C-NMR (CDCl_3_) δ 191.0, 133.7, 133.7, 128.7, 128.6, 30.7.

*2-Phenylimidazo[1,2-a]pyridine* (**3a**): white solid, ^1^H-NMR (CDCl_3_) δ 8.11 (dt, *J* = 6.8, 1.3 Hz, 1H), 8.00–7.91 (m, 2H), 7.85 (s, 1H), 7.64 (d, *J* = 9.1 Hz, 1H), 7.44 (t, *J* = 7.6 Hz, 2H), 7.37–7.30 (m, 1H), 7.17 (ddd, *J* = 9.1, 6.7, 1.3 Hz, 1H), 6.77 (td, *J* = 6.8, 1.2 Hz, 1H). ^13^C-NMR (CDCl_3_) δ 145.7, 145.6, 133.7, 128.7, 128.0, 126.0, 125.6, 124.7, 117.5, 112.4, 108.1. LRMS (ESI) *m/z*: [M + H]^+^ found 195.1, HRMS (ESI) *m/z*: [M + H]^+^ Calcd. for C_13_H_10_N_2_ 195.0917; found 195.0914.

*2-(4-Fluorophenyl)imidazo[1,2-a]pyridine* (**3b**): white solid, ^1^H-NMR (CDCl_3_) δ 8.11 (dd, *J* = 6.8, 1.3 Hz, 1H), 7.97–7.87 (m, 2H), 7.80 (s, 1H), 7.69–7.54 (m, 1H), 7.22–7.03 (m, 3H), 6.78 (td, *J* = 6.7, 1.2 Hz, 1H). ^13^C-NMR (CDCl_3_) δ 163.7, 161.7, 145.7, 144.9, 129.9, 127.7, 125.6, 124.8, 117.4, 115.7, 115.5, 112.5, 107.7. LRMS (ESI) *m/z*: [M + H]^+^ found 213.1, HRMS (ESI) *m/z*: [M + H]^+^ Calcd. for C_13_H_9_FN_2_ 213.0823; found 213.0822.

*2-(4-Bromophenyl)imidazo[1,2-a]pyridine* (**3c**): white solid, ^1^H-NMR (CDCl_3_) δ 8.10 (dt, *J* = 6.8, 1.2 Hz, 1H), 7.92–7.76 (m, 3H), 7.69–7.58 (m, 1H), 7.59–7.47 (m, 2H), 7.23–7.08 (m, 1H), 6.79 (td, *J* = 6.8, 1.2 Hz, 1H). ^13^C-NMR (CDCl_3_) δ 145.8, 144.8, 132.8, 132.0, 131.9, 127.7, 125.7, 125.0, 122.0, 117.7, 112.7, 108.3. LRMS (ESI) *m/z*: [M + H]^+^ found 273.1, HRMS (ESI) *m/z*: [M + H]^+^ Calcd. for C_13_H_10_BrN_2_ 273.0022; found 273.0017.

*2-(4-Chlorophenyl)imidazo[1,2-a]pyridine* (**3d**): white solid, ^1^H-NMR (CDCl_3_) δ 8.36 (d, *J* = 6.7 Hz, 1H), 8.07 (s, 1H), 8.03–7.92 (m, 3H), 7.53–7.46 (m, 1H), 7.43–7.37 (m, 2H), 7.10 (t, *J* = 6.8 Hz, 1H). ^13^C-NMR (CDCl_3_) δ 142.7, 135.5, 131.5, 129.3, 129.1, 128.2, 127.7, 126.5, 115.3, 108.8. LRMS (ESI) *m/z*: [M + H]^+^ found 229.0, HRMS (ESI) *m/z*: [M + H]^+^ Calcd. for C_13_H_9_ClN_2_ 229.0527; found 229.0527.

*2-(3,5-Difluorophenyl)imidazo[1,2-a]pyridine* (**3e**): white solid, ^1^H-NMR (CDCl_3_) δ 8.67 (s, 1H), 8.43 (s, 1H), 8.25 (d, *J* = 9.1 Hz, 1H), 7.70 (qt, *J* = 7.6, 4.5 Hz, 3H), 7.30 (d, *J* = 6.6 Hz, 1H), 6.86 (tt, *J* = 8.7, 2.3 Hz, 1H). ^13^C-NMR (CDCl_3_) δ 164.4, 164.3, 162.4, 162.3, 127.4, 116.8, 114.5, 110.0, 109.8, 105.8, 105.6, 105.4. LRMS (ESI) *m/z*: [M + H]^+^ found 231.0, HRMS (ESI) *m/z*: [M + H]^+^ Calcd. for C_13_H_8_F_2_N_2_ 231.0728 found 231.0725.

*4-(Imidazo[1,2-a]pyridin-2-yl)benzonitrile* (**3f**): white solid, ^1^H-NMR (CDCl_3_) δ 8.18 (dt, *J* = 6.7, 1.2 Hz, 1H), 8.11–8.07 (m, 2H), 7.99–7.96 (m, 1H), 7.77–7.68 (m, 3H), 7.29–7.23 (m, 1H), 6.88 (td, *J* = 6.8, 1.1 Hz, 1H). ^13^C-NMR (CDCl_3_) δ 145.7, 143.3, 137.9, 132.6, 126.4, 125.8, 125.8, 118.9, 117.7, 113.2, 111.2, 109.5. LRMS (ESI) *m/z*: [M + H]^+^ found 220.1, HRMS (ESI) *m/z*: [M + H]^+^ Calcd. for C_14_H_9_N_3_ 220.0869; found 220.0872.

*2-(4-(Trifluoromethyl)phenyl)imidazo[1,2-a]pyridine* (**3g**): white solid, ^1^H-NMR (CDCl_3_) δ 8.20 (dt, *J* = 6.8, 1.3 Hz, 1H), 8.11 (d, *J* = 8.1 Hz, 2H), 7.97 (s, 1H), 7.76 (d, *J* = 9.1 Hz, 1H), 7.72 (d, *J* = 8.2 Hz, 2H), 7.33–7.29 (m, 1H), 6.90 (td, *J* = 6.8, 1.1 Hz, 1H). ^13^C-NMR (CDCl_3_) δ 145.2, 143.3, 130.1, 129.9, 126.2, 126.0, 125.8, 125.8, 125.7, 125.2, 123.0, 117.4, 113.4, 109.0. LRMS (ESI) *m/z*: [M + H]^+^ found 263.0, HRMS (ESI) *m/z*: [M + H]^+^ Calcd. for C_14_H_9_F_3_N_2_ 263.0791; found 263.0791.

*2-(4-(Methylsulfonyl)phenyl)imidazo[1,2-a]pyridine* (**3h**): white solid, ^1^H-NMR (CDCl_3_) δ 8.19–8.11 (m, 3H), 8.04–7.94 (m, 3H), 7.64 (dd, *J* = 9.0, 1.1 Hz, 1H), 7.22 (ddd, *J* = 9.1, 6.7, 1.3 Hz, 1H), 6.83 (td, *J* = 6.8, 1.2 Hz, 1H), 3.08 (s, 3H). ^13^C-NMR (CDCl_3_) δ 145.7, 143.3, 139.0, 127.6, 126.3, 125.5, 125.2, 117.6, 112.8, 109.4, 44.3. LRMS (ESI) *m/z*: [M + H]^+^ found 272.9, HRMS (ESI) *m/z*: [M + H]^+^ Calcd. for C_14_H_12_N_2_O_2_S 273.0692; found 273.0694.

*2-(p-Tolyl)imidazo[1,2-a]pyridine* (**3i**): white solid, ^1^H-NMR (CDCl_3_) δ 8.23 (dt, *J* = 6.8, 1.2 Hz, 1H), 7.93–7.83 (m, 4H), 7.34–7.30 (m, 1H), 7.26 (d, *J* = 7.9 Hz, 2H), 6.96–6.88 (m, 1H), 2.40 (s, 3H). ^13^C-NMR (CDCl_3_) δ 144.1, 143.4, 138.8, 129.6, 126.61, 126.2, 125.9, 116.4, 113.7, 107.9, 21.3. LRMS (ESI) *m/z*: [M + H]^+^ found 209.1, HRMS (ESI) *m/z*: [M + H]^+^ Calcd. for C_14_H_12_N_2_ 209.1073; found 209.1072.

*2-(4-Methoxyphenyl)imidazo[1,2-a]pyridine* (**3j**): white solid, ^1^H-NMR (CDCl_3_) δ 8.15 (dt, *J* = 6.7, 1.2 Hz, 1H), 7.95–7.90 (m, 2H), 7.81 (d, *J* = 0.7 Hz, 1H), 7.74 (d, *J* = 9.0 Hz, 1H), 7.24 (ddd, *J* = 9.1, 6.8, 1.3 Hz, 1H), 7.02–6.97 (m, 2H), 6.84 (td, *J* = 6.8, 1.1 Hz, 1H), 3.87 (s, 3H). ^13^C-NMR (CDCl_3_) δ 159.9, 144.8, 144.5, 127.4, 125.6, 125.4, 125.2, 116.8, 114.2, 112.9, 107.2, 55.3. LRMS (ESI) *m/z*: [M + H]^+^ found 225.1, HRMS (ESI) *m/z*: [M + H]^+^ Calcd. for C_14_H_12_N_2_O 225.1022; found 225.1025.

*2-(Naphthalen-1-yl)imidazo[1,2-a]pyridine* (**3k**): white solid, ^1^H-NMR (CDCl_3_) δ 8.59–8.49 (m, 1H), 8.15 (dt, *J* = 6.7, 1.2 Hz, 1H), 8.01 (dd, *J* = 8.5, 1.7 Hz, 1H), 7.99 (d, *J* = 0.7 Hz, 1H), 7.96–7.88 (m, 2H), 7.86–7.82 (m, 1H), 7.69 (dd, *J* = 9.0, 1.2 Hz, 1H), 7.53–7.44 (m, 2H), 7.21 (ddd, *J* = 9.1, 6.7, 1.3 Hz, 1H), 6.81 (td, *J* = 6.7, 1.1 Hz, 1H). ^13^C-NMR (CDCl_3_) δ 145.7, 145.5, 140.7, 140.6, 132.7, 128.7, 127.4, 127.3, 126.9, 126.4, 125.5, 124.6, 117.5, 112.4, 108.2. LRMS (ESI) *m/z*: [M + H]^+^ found 244.9, HRMS (ESI) *m/z*: [M + H]^+^ Calcd. for C_17_H_12_N_2_ 245.1073; found 245.1072.

*8-Methyl-2-phenylimidazo[1,2-a]pyridine* (**3l**): white solid, ^1^H-NMR (CDCl_3_) δ 7.98–7.91 (m, 2H), 7.91–7.85 (m, 1H), 7.76 (s, 1H), 7.54 (d, *J* = 9.2 Hz, 1H), 7.42 (t, *J* = 7.7 Hz, 2H), 7.32 (d, *J* = 7.4 Hz, 1H), 7.01 (dd, *J* = 9.2, 1.7 Hz, 1H), 2.37–2.27 (m, 3H). ^13^C-NMR (CDCl_3_) δ 145.7, 145.1, 135.1, 133.5, 128.2, 127.4, 125.7, 124.3, 115.5, 114.6, 107.1, 21.0. LRMS (ESI) *m/z*: [M + H]^+^ found 209.2, HRMS (ESI) *m/z*: [M + H]^+^ Calcd. for C_14_H_12_N_2_ 209.1073; found 209.1075.

*7-Methyl-2-phenylimidazo[1,2-a]pyridine* (**3m**): white solid, ^1^H-NMR (CDCl_3_) δ 8.02–7.87 (m, 3H), 7.76 (s, 1H), 7.50–7.36 (m, 3H), 7.32 (d, *J* = 7.3 Hz, 1H), 6.59 (dd, *J* = 6.9, 1.7 Hz, 1H), 2.39 (d, *J* = 1.1 Hz, 3H). ^13^C-NMR (CDCl_3_) δ 145.7, 145.1, 135.2, 133.5, 128.2, 127.4, 125.6, 124.3, 115.5, 114.6, 107.1, 21.0. LRMS (ESI) *m/z*: [M + H]^+^ found 209.2, HRMS (ESI) *m/z*: [M + H]^+^ Calcd. for C_14_H_12_N_2_ 209.1073; found 209.1075.

*6-Methyl-2-phenylimidazo[1,2-a]pyridine* (**3n**): white solid, ^1^H-NMR (CDCl_3_) δ 8.00–7.86 (m, 3H), 7.77 (s, 1H), 7.54 (d, *J* = 9.2 Hz, 1H), 7.43 (t, *J* = 7.6 Hz, 2H), 7.36–7.29 (m, 1H), 7.01 (dd, *J* = 9.2, 1.7 Hz, 1H), 2.34–2.28 (m, 3H). ^13^C-NMR (CDCl_3_) δ 134.0, 128.8, 127.9, 127.9, 126.0, 123.4, 122.1, 116.9, 107.9, 18.1. LRMS (ESI) *m/z*: [M + H]^+^ found 209.2, HRMS (ESI) *m/z*: [M + H]^+^ Calcd. for C_14_H_12_N_2_ 209.1073; found 209.1075.

*6-Methoxy-2-phenylimidazo[1,2-a]pyridine* (**3o**): white solid, ^1^H-NMR (CDCl_3_) δ 7.95–7.87 (m, 2H), 7.79 (s, 1H), 7.63 (d, *J* = 2.4 Hz, 1H), 7.53 (d, *J* = 9.7 Hz, 1H), 7.42 (t, *J* = 7.7 Hz, 2H), 7.32 (d, *J* = 7.3 Hz, 1H), 6.97 (dd, *J* = 9.7, 2.4 Hz, 1H), 3.82 (s, 3H). ^13^C-NMR (CDCl_3_) δ 148.9, 145.2, 142.4, 133.5, 128.3, 127.3, 125.4, 119.4, 117.3, 108.8, 107.1, 55.8. LRMS (ESI) *m/z*: [M + H]^+^ found 225.1, HRMS (ESI) *m/z*: [M + H]^+^ Calcd. for C_14_H_12_N_2_O 225.1022; found 225.1018.

*5-Methoxy-2-phenylimidazo[1,2-a]pyridine* (**3p**): white solid, ^1^H-NMR (CDCl_3_) δ 7.97–7.92 (m, 2H), 7.83 (s, 1H), 7.68 (d, *J* = 2.4 Hz, 1H), 7.57 (d, *J* = 9.7 Hz, 1H), 7.45 (t, *J* = 7.8 Hz, 2H), 7.34 (td, *J* = 7.2, 1.3 Hz, 1H), 7.00 (dd, *J* = 9.7, 2.3 Hz, 1H), 3.86 (s, 3H). ^13^C-NMR (CDCl_3_) δ 149.4, 146.8, 145.1, 133.9, 128.6, 127.8, 126.0, 125.9, 109.4, 103.6, 87.7, 56.3. LRMS (ESI) *m/z*: [M + H]^+^ found 225.1, HRMS (ESI) *m/z:* [M + H]^+^ Calcd. for C_14_H_12_N_2_O 225.1022; found 225.1018

*6-Fluoro-2-phenylimidazo[1,2-a]pyridine* (**3q**): white solid, ^1^H-NMR (CDCl_3_) δ 8.20 (dd, *J* = 2.0, 0.9 Hz, 1H), 7.99–7.94 (m, 2H), 7.86 (d, *J* = 0.7 Hz, 1H), 7.62 (d, *J* = 9.5 Hz, 1H), 7.47 (dd, *J* = 8.4, 7.0 Hz, 2H), 7.41–7.35 (m, 1H), 7.18 (dd, *J* = 9.5, 2.0 Hz, 1H). ^13^C-NMR (CDCl_3_) δ 146.6, 144.0, 133.1, 128.8, 128.3, 128.1, 126.1, 125.5, 118.1, 108.2, 107.0. LRMS (ESI) *m/z*: [M + H]^+^ found 213.2, HRMS (ESI) *m*/*z*: [M + H]^+^ Calcd. for C_13_H_9_FN_2_ 213.0823; found 213.0822.

*6-Chloro-2-phenylimidazo[1,2-a]pyridine* (**3r**): white solid, ^1^H-NMR (CDCl_3_) δ 8.18 (dd, *J* = 2.0, 0.9 Hz, 1H), 8.03–7.88 (m, 2H), 7.84 (s, 1H), 7.60 (d, *J* = 9.5 Hz, 1H), 7.45 (t, *J* = 7.5 Hz, 2H), 7.36 (d, *J* = 7.3 Hz, 1H), 7.15 (dd, *J* = 9.5, 2.0 Hz, 1H). ^13^C-NMR (CDCl_3_) δ 147.1, 144.3, 133.6, 129.1, 128.6, 126.4, 126.4, 123.7, 120.9, 118.2, 108.8. LRMS (ESI) *m/z*: [M + H]^+^ found 229.1, HRMS (ESI) *m/z*: [M + H]^+^ Calcd. for C_13_H_9_ClN_2_ 229.0527; found 229.0526.

*6-Bromo-2-phenylimidazo[1,2-a]pyridine* (**3s**): white solid, ^1^H-NMR (CDCl_3_) δ 8.33–8.21 (m, 1H), 8.00–7.89 (m, 2H), 7.82 (s, 1H), 7.53 (d, *J* = 9.5 Hz, 1H), 7.44 (dd, *J* = 8.3, 6.8 Hz, 2H), 7.39–7.30 (m, 1H), 7.23 (dd, *J* = 9.5, 1.9 Hz, 1H). ^13^C-NMR (CDCl_3_) δ 147.0, 144.4, 133.5, 129.1, 128.6, 128.4, 126.4, 125.9, 118.5, 108.6, 107.3. LRMS (ESI) *m/z*: [M + H]^+^ found 273.1, HRMS (ESI) *m/z*: [M + H]^+^ Calcd. for C_13_H_10_BrN_2_ 273.0022; found 273.0017.

*2-Phenyl-6-(trifluoromethyl)imidazo[1,2-a]pyridine* (**3t**): white solid, ^1^H-NMR (CDCl_3_) δ 8.25 (d, *J* = 7.0 Hz, 1H), 8.02–7.96 (m, 4H), 7.49 (t, *J* = 7.7 Hz, 2H), 7.43–7.37 (m, 1H), 6.99 (dd, *J* = 7.1, 1.8 Hz, 1H). ^13^C-NMR (CDCl_3_) δ 147.8, 145.3, 132.9, 128.9, 128.6, 126.2, 124.6, 120.6, 118.1, 109.2. LRMS (ESI) *m/z*: [M + H]^+^ found 263.1, HRMS (ESI) *m/z*: [M + H]^+^ Calcd. for C_14_H_10_ F_3_N_2_ 263.0791; found 263.0794.

*2-Phenylimidazo[1,2-a]pyridine-6-carbonitrile* (**3u**): white solid, ^1^H-NMR (CDCl_3_) δ 8.20 (dd, *J* = 7.0, 1.0 Hz, 1H), 8.08–7.90 (m, 4H), 7.46 (d, *J* = 7.8 Hz, 2H), 7.40 (d, *J* = 7.3 Hz, 1H), 6.94 (dd, *J* = 7.0, 1.6 Hz, 1H). ^13^C-NMR (CDCl_3_) δ 149.0, 143.5, 132.6, 129.0, 126.3, 126.2, 123.7, 117.7, 112.8, 110.3, 107.2. LRMS (ESI) *m/z*: [M + H]^+^ found 220.1, HRMS (ESI) *m/z*: [M + H]^+^ Calcd. for C_14_H_9_N_3_ 220.0869; found 220.0862.

*2-Phenylbenzo[d]imidazo[2,1-b]thiazole* (**5a**): white solid, ^1^H-NMR (CDCl_3_) δ 7.95 (d, *J* = 2.1 Hz, 1H), 7.91–7.82 (m, 2H), 7.72–7.63 (m, 1H), 7.63–7.55 (m, 1H), 7.45–7.38 (m, 3H), 7.36–7.29 (m, 2H). ^13^C-NMR (CDCl_3_) δ 148.1, 147.7, 133.8, 132.2, 130.3, 128.7, 127.5, 126.2, 125.2, 124.9, 124.4, 112.6, 106.8. LRMS (ESI) *m/z*: [M + H]^+^ found 251.1, HRMS (ESI) *m/z*: [M + H]^+^ Calcd. for C_15_H_10_N_2_S 251.0637; found 251.0632.

*6-Phenylimidazo[2,1-b]thiazole* (**5b**): white solid, ^1^H-NMR (CDCl_3_) δ 7.88–7.82 (m, 2H), 7.77 (s, 1H), 7.47–7.39 (m, 3H), 7.34–7.28 (m, 1H), 6.85 (d, *J* = 4.5 Hz, 1H). ^13^C-NMR (CDCl_3_) δ 150.2, 147.9, 134.1, 128.6, 127.3, 125.2, 118.4, 112.4, 107.9. LRMS (ESI) *m/z*: [M + H]^+^ found 201.1, HRMS (ESI) *m/z*: [M + H]^+^ Calcd. for C_11_H_9_N_2_S 201.0481; found 201.0482.

*N,4-Diphenylthiazol-2-amine* (**5c**): white solid, ^1^H-NMR (CDCl_3_) δ 7.88 (dd, *J* = 8.2, 1.4 Hz, 2H), 7.49–7.30 (m, 7H), 7.09 (tt, *J* = 6.9, 1.5 Hz, 1H), 6.84 (s, 1H). ^13^C-NMR (CDCl_3_) δ 164.3, 139.7, 133.9, 129.0, 128.2, 127.5, 125.7, 122.6, 117.8, 101.2. LRMS (ESI) *m/z*: [M + H]^+^ found 253.0, HRMS (ESI) *m/z*: [M + H]^+^ Calcd. for C_15_H_12_N_2_S 253.0794; found 253.0797.

*4-Phenylthiazol-2-amine* (**5d**): white solid, ^1^H-NMR (CDCl_3_) δ 7.86–7.67 (m, 2H), 7.38 (t, *J* = 7.5 Hz, 2H), 7.30 (d, *J* = 7.4 Hz, 1H), 6.72 (s, 1H), 5.26 (s, 2H). ^13^C-NMR (CDCl_3_) δ 167.3, 151.3, 134.7, 128.5, 127.7, 126.0, 102.8. LRMS (ESI) *m/z*: [M + H]^+^ found 177.1, HRMS (ESI) *m/z*: [M + H]^+^ Calcd. for C_9_H_8_N_2_S 177.0481; found 177.0486.

*2-Methyl-4-phenylthiazole* (**5e**): white solid, ^1^H-NMR (CDCl_3_) δ 7.93–7.88 (m, 2H), 7.44 (dd, *J* = 8.3, 7.0 Hz, 2H), 7.38–7.31 (m, 2H), 2.81 (s, 3H). ^13^C-NMR (CDCl_3_) δ 165.9, 155.1, 134.5, 128.7, 128.0, 126.3, 112.2, 19.3. LRMS (ESI) *m/z*: [M + H]^+^ found 176.0, HRMS (ESI) *m/z*: [M + H]^+^ Calcd. for C_10_H_9_NS 176.0534; found 176.0530.

*2,4-Diphenylthiazole* (**5f**): white solid, ^1^H-NMR (CDCl_3_) δ 8.09 (dd, *J* = 7.8, 1.7 Hz, 2H), 8.06–8.01 (m, 2H), 7.56–7.44 (m, 6H), 7.43–7.34 (m, 1H). ^13^C-NMR (CDCl_3_) δ 168.0, 133.6, 130.1, 128.9, 128.8, 128.2, 126.7, 126.5, 112.6. LRMS (ESI) *m/z*: [M + H]^+^ found 238.0, HRMS (ESI) *m/z*: [M + H]^+^ Calcd. for C_15_H_11_NS 238.0612; found 238.0614.

*2-Phenylimidazo[2,1-a]isoquinoline* (**5g**): white solid, ^1^H-NMR (CDCl_3_) δ 8.77 (d, *J* = 8.0 Hz, 1H), 8.12–8.01 (m, 2H), 7.90 (d, *J* = 7.2 Hz, 1H), 7.83 (s, 1H), 7.73–7.70 (m, 1H), 7.67 (ddd, *J* = 8.2, 7.1, 1.2 Hz, 1H), 7.59 (ddd, *J* = 8.2, 7.2, 1.3 Hz, 1H), 7.48 (t, *J* = 7.7 Hz, 2H), 7.40–7.33 (m, 1H), 7.05 (d, *J* = 7.2 Hz, 1H). ^13^C-NMR (CDCl_3_) δ 144.0, 143.3, 134.0, 129.4, 128.7, 128.1, 127.5, 126.9, 125.8, 123.8, 123.5, 122.9, 113.0, 109.8. LRMS (ESI) *m/z*: [M + H]^+^ found 244.9, HRMS (ESI) *m/z*: [M + H]^+^ Calcd. for C_17_H_12_N_2_ 245.1073; found 245.1072.

## 4. Conclusions

In summary, we demonstrated a one-pot approach for the synthesis of various imidazoles and thiazoles. The process involves the reaction of different ethylarenes with different suitable nucleophiles via treatment with NBS and a catalytic amount of AIBN in a mixture of ethyl acetate and water, to give corresponding products. It is notable that all reactions were carried out in water as the solvent and were metal-free, with NBS playing a dual role as both a bromine source and oxidant.

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
