# Peer review of "One-Pot NBS-Promoted Synthesis of Imidazoles and Thiazoles from Ethylarenes in Water"

_molecules, 2019, doi:10.3390/molecules24050893_

Round 1

Reviewer 1 Report

Wang, Liu and co-workers describe in this paper their preparative procedure for making imidazopyridines and related heterocycles.   The starting materials are ethylarenes and these are treated in turn with NBS and AIBN in EA/water and then a 2-amino-heterocycle and sodium carbonate.   They carried out a sequence of experiments to optimize the experimental conditions and then showed the process works well for arenes with electron-withdrawing as well as electron-releasing substituents.   They extended the scope to show that a variety of other heterocycles could be accessed by use of other 2-amino-heterocycles.  Yields were all good to very good.   They also performed several experiments designed to show up the mechanism. They then proposed a mechanism that seems reasonable and fits the known facts.  As far as I can tell the experiments were performed to a good professional standard and the products appear to have been satisfactorily characterized.   The research has an element of novelty, it is certainly a relatively facile route to the products, and may be milder than alternatives.   I recommend publication subject to the following points being rectified.

1. Page 1, line 31:  correct “the use of mental and” to: the use of metal and

2. Page 1, line 34: change “the reaction need expensive” to: the reaction needs expensive

3. Page 3, line 69 and in other places the authors refer to “acetic ether”.  I understand this is actually ethyl acetate.   Chemists do not use “acetic ether” because the compound is not an ether. Elsewhere in the paper Wang et al. use “ethyl acetate” so they should be consistent and change all the “acetic ether” to ethyl acetate.

4. Page 6, line 116: In the text 3a is said to have been prepared on a gram scale of 1.08 g.   However, in Scheme 2, product 3a is shown with a yield of 2.95 g!   There seems to be contradiction here!

4. Page 6, line 119, 120: The authors use the expression “was got” and “was get” (also on page 7, line 128 - and possibly elsewhere).  This is poor grammar.   ‘Was obtained’ is better.

5. Page 7:  In a number of places compound 1ab is called “alpha,alpha-bibromoethylbenzene”.   “Bibromo” is an old-fashioned terminology.  It is better to use ‘dibromo’ – and in fact the authors do call the same compound alpha,alpha-dibromoethylbenzene on line 146.  They should be consistent!

6. Page 9, line 184: NMR data is given for 2-bromo-1-phenylethan-1-one (1aa).  It appears this is not a new compound so a reference should be included to the literature preparation.

Author Response

Question 1: Page 1, line 31: correct “the use of mental and” to: the use of metal and.

Response: Thanks a lot for your corrections.

We have changed “the use of mental and” to “the use of metal andin our revised manuscript.

Question 2: Page 1, line 34: change “the reaction need expensive” to: the reaction needs expensive

Response: Thanks a lot for your corrections.

We have changed “the reaction need expensive” to “the reaction needs expensive” in our revised manuscript.

Question 3: Page 3, line 69 and in other places the authors refer to “acetic ether”. I understand this is actually ethyl acetate. Chemists do not use “acetic ether” because the compound is not an ether. Elsewhere in the paper Wang et al. use “ethyl acetate” so they should be consistent and change all the “acetic ether” to ethyl acetate.

Response: Thanks a lot for your corrections.

We have changed “acetic ether” to “ethyl acetate” throughout the revised manuscript.

Question 4: Page 6, line 116: In the text 3a is said to have been prepared on a gram scale of 1.08 g. However, in Scheme 2, product 3a is shown with a yield of 2.95 g!   There seems to be contradiction here!

Response: Thanks a lot for your corrections.

We have changed “preparation of 3a on a gram scale (1.08 g)” to “preparation of 3a on a gram scale (2.98 g)” in our revised manuscript.

Question 5: Page 6, line 119, 120: The authors use the expression “was got” and “was get” (also on page 7, line 128 and possibly elsewhere).  This is poor grammar. ‘Was obtained’ is better.

Response: Thanks a lot for your corrections.

We have changed “was got” to “was obtained” and “was get” to “was obtained” in our revised manuscript.

Question 6: Page 7:  In a number of places compound 1ab is called “alpha,alpha-bibromoethylbenzene”. “Bibromo” is an old-fashioned terminology.  It is better to use ‘dibromo’ – and in fact the authors do call the same compound alpha,alpha-dibromoethylbenzene on line 146.  They should be consistent!

Response: Thanks a lot for your corrections.

We have changed “α,α-dibromoethybenzene” to “α,α-dibromoethybenzene” throughout the revised manuscript.

Question 7: Page 9, line 184: NMR data is given for 2-bromo-1-phenylethan-1-one (1aa).  It appears this is not a new compound so a reference should be included to the literature preparation.

Response: Thanks a lot for your suggestions.

We have added literature cited as reference 42 in our revised manuscript .

“42. Gonzalez-de-Castro, A.; Xiao, J., Green and Efficient: Iron-Catalyzed Selective Oxidation of Olefins to Carbonyls with O2. J. Am. Chem. Soc. 2015, 137, (25), 8206-8218."

Reviewer 2 Report

Nice research project aiming at the synthesis of imidazoazine/azole fused systems out of simplified precursors. I believe this study is consistent, carefully done and deserves publication.

However, there are some points to be addressed before approval, to improve quality and suitability for potential readers.

1- Title- Is is too general and inclonclusive. Better restrict to the imidazofused N-systems.

2- Please, correct the following sentence (page 1, line 31) "Traditionally, procedures for these reductions required the use of mental and catalyst in the presence various organic solvents..." Should be reaction, not reduction, metal, not mental.

(also check the whole text, as I may have overlooked some more mistakes).

3- Check in databases if the synthesis of alpha-haloketones has been done in a similar way, from ethylarenes and NXS. If so, state it clearly.

4- Plase, add the literature references for the knowndescribed compounds 3.

5- Please, add the 1H and 13C NMR spectra for the synthesized compounds in the supporting information section.

Author Response

Question 1: Title- Is is too general and inclonclusive. Better restrict to the imidazofused N-systems.

Response: Thanks a lot for your suggestions.

We have changed the title to “One-pot synthesis of imidazoles and thiazoles from ethylarenes in water promoted by NBS” in our revised manuscript.

Question 2: Please, correct the following sentence (page 1, line 31) "Traditionally, procedures for these reductions required the use of mental and catalyst in the presence various organic solvents..." Should be reaction, not reduction, metal, not mental.

(also check the whole text, as I may have overlooked some more mistakes).

Response: Thanks a lot for your corrections.

We have changed “Traditionally, procedures for these reductions required the use of mental and catalyst in the presence various organic solvents..." to “Traditionally, procedures for these reactions required the use of metal and catalyst in the presence various organic solvents...”.

Furthermore, we have carefully checked the whole text and made corrections listed as below:

1.     We have changed “the reaction need expensive” to “the reaction needs expensive” in our revised manuscript.

2.     We have changed “acetic ether” to “ethyl acetate” throughout the revised manuscript.

3.     We have changed “preparation of 3a on a gram scale (1.08 g)” to “preparation of 3a on a gram scale (2.98 g)” in our revised manuscript.

4.     We have changed “was got” to “was obtained” and “was get” to “was obtained” in our revised manuscript.

5.     We have changed “α,α-dibromoethybenzene” to “α,α-dibromoethybenzene” throughout the revised manuscript.

6.     We have changed “Shohei et al” to “Shimokawa et al” in our revised manuscript.

Question 3: Check in databases if the synthesis of alpha-haloketones has been done in a similar way, from ethylarenes and NXS. If so, state it clearly.

Response: Thanks a lot for your comments.

We have checked in databases. Shimokawa et al.repoted the synthesis of alpha-bromoketone from ethylarenes and NBS in a similar way. This literature has been cited as reference 31 in our manuscript.

31. Shimokawa, S.; Kawagoe, Y.; Moriyama, K.; Togo, H., Direct Transformation of Ethylarenes into Primary Aromatic Amides with N-Bromosuccinimide and I2-Aqueous NH3. Org. Lett. 2016, 18, (4), 784-787.

Question 4: Please, add the literature references for the known described compounds 3

Response: Thanks a lot for your comments.

We have added literature references 10-16 listed as below for the known described compounds 3 in our revised manuscript.

Detail references:

10.  Gunaganti, N.; Kharbanda, A.; Lakkaniga, N. R.; Zhang, L.; Cooper, R.; Li, H. Y.; Frett, B., Catalyst free, C-3 functionalization of imidazo[1,2-a]pyridines to rapidly access new chemical space for drug discovery efforts. Chem. Commun. 2018, 54, (92), 12954-12957.

11.  Guo, T.; Liang, J. J.; Yang, S.; Chen, H.; Fu, Y. N.; Han, S. L.; Zhao, Y. H., Palladium-catalyzed oxidative C-H/C-H cross-coupling of imidazopyridines with azoles. Org. Biomol. Chem. 2018, 16, (33), 6039-6046.

12.  Guo, Y. J.; Lu, S.; Tian, L. L.; Huang, E. L.; Hao, X. Q.; Zhu, X.; Shao, T.; Song, M. P., Iodine-Mediated Difunctionalization of Imidazopyridines with Sodium Sulfinates: Synthesis of Sulfones and Sulfides. J. Org. Chem. 2018, 83, (1), 338-349.

13.  Kitson, P. J.; Marie, G.; Francoia, J. P.; Zalesskiy, S. S.; Sigerson, R. C.; Mathieson, J. S.; Cronin, L., Digitization of multistep organic synthesis in reactionware for on-demand pharmaceuticals. Science. 2018, 359, (6373), 314-319.

14.  Lefin, R.; van der Walt, M. M.; Milne, P. J.; Terre'Blanche, G., Imidazo[1,2-alpha]pyridines possess adenosine A1 receptor affinity for the potential treatment of cognition in neurological disorders. Bioorg. Med. Chem. Lett. 2017, 27, (17), 3963-3967.

15.  McDonald, I. M.; Peese, K. M., General Method for the Preparation of Electron-Deficient Imidazo[1,2-a]pyridines and Related Heterocycles. Org. Lett. 2015, 17, (24), 6002-6005.

16.  Sun, K.; Mu, S.; Liu, Z.; Feng, R.; Li, Y.; Pang, K.; Zhang, B., Copper-catalyzed C-N bond formation with imidazo[1,2-a]pyridines. Org. Biomol. Chem. 2018, 16, (36), 6655-6658.

Question 5: Please, add the 1H and 13C NMR spectra for the synthesized compounds in the supporting information section.

Response: Thanks a lot for your comments.

We have uploaded the supporting information files containing 1H and 13C NMR spectra of all the synthesized compounds.

Reviewer 3 Report

The authors claimed that one-pot synthesis of heterocycles from ethylarenes with NBS-AIBN systems.

Their results are very useful for many readers to synthesize nitrogen containing heterocycles.

There are several unclear points in the manuscript, so reviewer recommends their submission will be accepted after checking the revised manuscript.

Detailed are listed below.

1)   Lane 49 “Shohei et al” for ref 24, but Lane 386 for ref 24 is “Shimokawa, S.; Kawagoe, Y.; Moriyama, K.; Togo, H. “

2)   Lane 71 What kind of the reactions did detect for NCS and NIS in 1ststep ? It is not difficult for authors because authors can identify the products for NBS on 1ststep in Scheme 3. 

3)   Lane 75 “acetic ether” ?  

4)   Lane 76 Reviewer cannot access “ESI” on the reviewer’ HP. Did authors submit “ESI” for their submission? 

5)    Author must provide more examples for substitution effects on benzene ring (compounds 1 in Table 1) in the reactions. Authors afforded the results only 4 samples, p-F and p-Br benzene, naphthalene, and compound 6. Many substituted benzene derivatives have been reported in “recent methods in Scheme 1”, which utilized NBS as reactants. AIBN effects for the substitutions on benzene is quite important for their methods. Furthermore, picked many biologically active compounds in Figure 1 contained several substitutions on benzene ring.  

Author Response

Question 1: Line 49 “Shohei et al” for ref 24, but Line 386 for ref 24 is “Shimokawa, S.; Kawagoe, Y.; Moriyama, K.; Togo, H.

Response: Thanks a lot for your corrections.

We have changed “Shohei et al” to “Shimokawa et al” in our revised manuscript.

Question 2: Line 71 What kind of the reactions did detect for NCS and NIS in 1ststep ? It is not difficult for authors because authors can identify the products for NBS on 1ststep in Scheme 3

Response: Thanks a lot for your comments.

Only trace of corresponding alpha-haloketones were detected in 1st step using NCS and NIS.

Question 3: Line 75 “acetic ether” ? 

Response: Thanks a lot for your corrections.

We have changed “acetic ether” to “ethyl acetate” throughout the revised manuscript.

Question 4: Line 76 Reviewer cannot access “ESI” on the reviewer’ HP. Did authors submit “ESI” for their submission?

Response: Thanks a lot for your comments.

We have uploaded the supporting information files.

Question 5: Author must provide more examples for substitution effects on benzene ring (compounds 1 in Table 1) in the reactions. Authors afforded the results only 4 samples, p-F and p-Br benzene, naphthalene, and compound 6. Many substituted benzene derivatives have been reported in “recent methods in Scheme 1”, which utilized NBS as reactants. AIBN effects for the substitutions on benzene is quite important for their methods. Furthermore, picked many biologically active compounds in Figure 1 contained several substitutions on benzene ring. 

Response: Thanks a lot for your comments. We have added another six examples containing different substituent group on the benzene ring ( yields 32%-68%, 3d-3g,3i and 3j). Moreover, compound 3h is Zolimidine, which is a gastroprotective drug previously used for peptic ulcer and gastroesophageal reflux disease. And, compounds 3a-3d, 3g, 3i and 3j display adenosine A1 receptor binding affinity (see reference 14).

Revised “2.2. Substrate scope for the imidazo[1,2-a]pyridines…” are shown as below:

2.2. Substrate scope for the imidazo[1,2-a]pyridines

Having optimized conditions in hand, the reactions of different 2-aminopyridines with various ethylarenes were investigated to explore the scope and generality of this method for the synthesis of various imidazo[1,2-a]pyridines (Table 2). The reaction of 2-aminopyridine 2a with ethylarenes containing electron-withdrawing groups on the aromatic ring such as halogen, trifluoromethyl, methylsulfonyl, and cyano, provided the corresponding products (3b-3h) smoothly in medium to good yields (32% and 65%). Ethylbenzene with electron-rich groups, such as methyl and methoxy, also afforded the corresponding imidazo[1,2-a]pyridine (3i and 3j) in very good yield (68% and 59%). The same treatment of 1-ethylnaphthalene gave desired product 3k in good yield (65%). Furthermore, electron-rich and electron-withdrawing groups of 2-aminopyridines were well tolerated. The result revealed that electron-rich groups such as methyl and methoxyl, afforded the corresponding products in very good yield (yields 75%-80%, 3l-3p). However, introduction of the electron-withdrawing groups such as halogen, trifluoromethyl, and cyano afford corresponding products in lower yields (yields 56%-71%, 3q-3u).

Round 2

Reviewer 3 Report

Undesired signals between 0.5~4.5 ppm were detected in 1H-NMR for compound 3k, 3p, 3t, 5b and 5e in SI.

It cannot imagine what kind of impurities contaminated in their reactions.

Author Response

Response: Thanks a lot for your comments.

These five compounds mentioned above have been synthesized and purified again. 1H-NMR spectra for these five compounds are shown as below. In addition, the manuscript and supporting information have been revised.
